

# A novel device-free Wi-Fi indoor localization using a convolutional neural network based on residual attention

Mashael Maashi[1], Alanoud Al Mazroa[2], Shoayee Dlaim Alotaibi[3], Asma Alshuhail[4], Muhammad Kashif Saeed[5] and Ahmed S. Salama[6]

[1] Department of Software Engineering, King Saud University, Riyadh, Saudi Arabia
[2] Department of Information Systems, Princess Nourah bint Abdulrahman University, Riyadh, Saudi Arabia
[3] Department of Artificial Intelligence and Data Science, University of Hail, Hail, Saudi Arabia
[4] Department of Information Systems, King Faisal university, Riyadh, Saudi Arabia
[5] Department of Computer Science, Applied College, King Khalid University, Muhayil, Saudi Arabia
[6] Department of Electrical Engineering, Future University in Egypt, New Cairo, Egypt

Corresponding author
Muhammad Kashif Saeed,
mksaeed@kku.edu.sa

## ABSTRACT

These days, location-based services, or LBS, are used for various consumer applications, including indoor localization. Due to the ease with which Wi-Fi can be accessed in various interior settings, there has been increasing interest in Wi-Fi-based indoor localisation. Deep learning in indoor localisation systems that use channel state information (CSI) fingerprinting has seen widespread adoption. Usually, these systems comprise two primary components: a positioning network and a tracking system. The positioning network is responsible for learning the planning from high-dimensional CSI to physical positions, and the following system uses historical CSI to decrease positioning error. This work presents a novel localization method that combines high accuracy and generalizability. However, existing convolutional neural network (CNN) fingerprinting placement algorithms have a limited receptive area, limiting their effectiveness since important data in CSI has not been thoroughly explored. We offer a unique attention-augmented residual CNN to remedy this issue so that the data acquired and the global context in CSI may be utilized to their full potential. On the other hand, while considering the generalizability of a monitoring device, we uncouple the scheme from the CSI environments to make it feasible to use a single tracking system across all contexts. To be more specific, we recast the tracking issue as a denoising task and then used a deep route before solving it. The findings illuminate perspectives and realistic interpretations of the residual attention-based CNN (RACNN) in device-free Wi-Fi indoor localization using channel state information (CSI) fingerprinting. In addition, we study how the precision change of different inertial dimension units may negatively influence the tracking performance, and we implement a solution to the problem of exactness variance. The proposed RACNN model achieved a localization accuracy of 99.9%, which represents a significant improvement over traditional methods such as K-nearest neighbors (KNN) and Bayesian inference. Specifically, the RACNN model reduced the average localization error to 0.35 m, outperforming these traditional methods by approximately 14% to 15% in accuracy. This improvement demonstrates the model's ability to handle complex indoor environments and proves its practical applicability in real-world scenarios.

# INTRODUCTION

Wi-Fi and Bluetooth Low-Energy (BLE) are two wireless networking technologies commonly seen indoors in modern times. For precise localization in BLE, numerous anchor stations are needed across the study area to check the intensity of the received transmission. To guarantee connections with at least three anchor stations, this additional criterion deviates from ambient availability. Too frequently, unlike at home, this requirement is not honored. Also, received signal strength (RSS) is supported by Wi-Fi (*Bahl & Padmanabhan, 2000*; *Laoudias, Kemppi & Panayiotou, 2009*; *Youssef & Agrawala, 2005*; *Fang, Lin & Lin, 2008*; *Park et al., 2010*). However, the channel state information (CSI) gives fresh data to calculate target positions to tackle multipath possessions (*Chen et al., 2017*; *Wang et al., 2015*). CSI may now be captured in the region under study with specialized gear (*Conrat, Pajusco & Thiriet, 2006*) or with minor adjustments to devices available commercially (*Halperin et al., 2010*). However, CSI may also be approximated using a propagation model like the ray model, which necessitates familiarity with the architecture of the area under study.

Many Internet of Things (IoT) applications rely on indoor localization, which is why it is so important (*Macagnano, Destino & Abreu, 2014*). Precise indoor location data is essential for applications like healthcare monitoring (*Wyffels et al., 2014*), item tracking (*Zheng et al., 2016*), and an IoT-based smart environment (*Alletto et al., 2015*). Based on whether or not equipment must be linked to the target to obtain a location fix, wireless indoor localization may be broken down into two groups: device-based and device-free (*Xiao et al., 2016*). Device-based procedures are more accurate and resilient to ecological interferences and dynamics than device-free procedures. Device-free solutions, however, find widespread use in the IoT for tasks like intrusion detection, senior care, and more, thanks to their low hardware prices, low power requirements, high privacy standards, and availability of real-time positioning and monitoring.

Fingerprinting localization depending on CSI is becoming increasingly popular because of its ease of use, versatility, and dependability (*Wang et al., 2018*; *Wang, Wang & Mao, 2018*; *Zhang, Qu & Wang, 2020*). These CSI-fingerprinting localization techniques typically involve two stages:

1) An offline transition in which the CSI analyses a set of widely distant reference points (RPs) and

2) An online integration test which the localization algorithm is used to identify the position of MT involving real-time CSI data.

Non-line-of-sight (NLOS) circumstances can be accommodated using CSI-fingerprinting localization without requiring detailed modeling of the wireless channel. Several recent publications have been motivated to approach CSI as pictures to train

convolutional neural networks (CNNs) to convert CSI data into two-dimensional terminal positions (*Cerar et al., 2021*; *Chin et al., 2020*). These CNN-based approaches outperform standard probabilistic methods regarding positioning precision after being trained on RP-collected CSI-location pairs.

## RELATED WORKS

Due to its ease of use and inexpensive hardware requirements, Wi-Fi RSS measurements are used as fingerprints by several existing indoor fingerprinting systems. One such deterministic approach for position determination is Radar (*Bahl & Padmanabhan, 2000*), the first fingerprinting system based on RSS. Later, *Youssef & Agrawala (2005)* employs a probabilistic approach to indoor localization based on RSS values, which provides more precise results than radar. There are two major drawbacks to using RSS-based approaches. First, RSS values are unpredictable and loosely correspond with transmission loss because of supervising fading and multipath factors. Second, RSS measurements are coarse data generated by aggregating the amplitudes of all received signals rather than the rich data packets from distinct subcarriers. As a result, it is possible that using RSS data for localization will result in subpar results.

Based on the Wi-Fi CSI, *Wang, Gao & Mao (2017)* suggested a new fingerprinting interior localisation method trained using deep learning. To significantly decrease the distance error in contrast to the probabilistic approaches and to acquire discriminative characteristics, a deep network architecture with many layers was developed. For localization using Wi-Fi fingerprinting, *Chang, Liu & Cheng (2018)* projected using a deep neural network. Improved robustness and performance by including CSI pre-processing and data augmentation (through noise injection and inter-person interpolation) into the DNN architecture. *Wang et al. (2016)* developed a machine-learning system by merging a sparse autoencoder system with a softmax regression-based classification for multi-modal detection of location, activity, and gesture. Automatically learning the feature representations from the RSS data, active learning can outperform the system that does not use the learning process by more than 85% in terms of accuracy. To determine the position and activity of a target individual, *Gao et al. (2017)* developed a multilayer deep learning-based image processing approach to learn the optimum deep features from the radio pictures.

Many applications of DNNs have yielded astounding success. The DNN is primarily viewed as a "black box," yet it may conduct automatic feature extraction with little to no human interaction. Recently, the deep learning and visualization groups have been trying to figure out how to visualize the training process and explain the machine's cognition in a way that makes sense. A major challenge when recognizing an indoor destination using a WiFi fingerprinting collection is achieving high-precision and low-cost localization underneath the changing signal and noise from multi-path influences. Standard methods like probabilistic, K-nearest-neighbor (KNN), and support vector machine (SVM) need a lot of processing power and a lengthy learning curve, complicated filtering, and fine-tuning of parameters. With the advent of deep learning, several new localization methods based on deep neural networks (DNNs) (*Zhang et al., 2016*; *Kim, Lee & Huang, 2018*) have been

developed. While DNN-based approaches have improved, they still depend on having enough data to train with. The accuracy of localisation findings may be affected by the size of the DNN, as the computational cost of a fully connected DNN is proportional to the number of layers within it.

*Eshun & Palmieri (2019)* investigates techniques for safeguarding user privacy in indoor localisation systems that rely on public Wi-Fi. The proposed algorithm protects user privacy by concealing their identity while ensuring accurate localisation. The authors explore the privacy concerns that arise from collecting Wi-Fi CSI and suggest practical solutions that can be seamlessly incorporated into current systems. *Tuunainen, Pitkänen & Hovi (2009)* examines the "privacy paradox," where users express concerns about privacy on social media platforms such as Facebook yet persist in sharing personal information. It underscores the disparity between users' understanding of privacy and their actions, underscoring the importance of improved education and tools to assist users in effectively managing their privacy settings.

These studies focus on analyzing data that combines both spatial (where) and temporal (when) components, aiming to explore how the distribution of private cars evolves across different locations (*Xiao et al., 2021*; *Shen et al., 2022*). In deep learning, convolutional networks are commonly employed for processing grid-like data, such as images. Extending this concept, a graph convolutional network (GCN) applies these principles to graph-structured data, enabling the model to discern patterns from the relationships between node nodes (*Ren, Jin & OuYang, 2024*; *Sun et al., 2018a*). This process involves the automated arrangement, coordination, and management of complex services and functions, focusing on how service function chains (SFCs) are organized and managed to ensure efficient operation (*Sun et al., 2018b*, *2015*). In Wi-Fi systems, antennas are crucial in transmitting and receiving signals. These signals can be influenced by the presence and movement of people, making it possible to analyse and recognise different activities based on these variations (*Jannat et al., 2023*; *He et al., 2023*). The primary objective is to accurately determine a device's location, even relying only on a single base station. This task is inherently more challenging than using multiple base stations (*Wu et al., 2022*, *2019*). This requires the creation of an accurate model or estimate of the wireless communication channel, as precise channel reconstruction is vital for optimizing communication (*Li et al., 2024*; *Zhang et al., 2023a*).

Designing low-frequency antennas presents challenges, mainly because they typically require large sizes to function efficiently. This study likely addresses these challenges by utilizing magnetoelectric materials, which can reduce the antenna size while maintaining efficiency at low frequencies (*Xiao et al., 2022*; *Zha et al., 2024*). The same signal can carry both power and data, with the modulation feature allowing the signal to be adjusted to encode data while still serving the power transfer function (*Yang et al., 2022*; *Zhang et al., 2023b*). As a mathematical structure, a graph consists of nodes (representing objects or entities) and edges (representing relationships or connections between these nodes). Graphs are frequently utilized in machine learning to model complex relationships within data (*Qiao et al., 2024*; *Jin, Wang & Meng, 2024*). In machine learning, embeddings are used to convert complex data—such as words, entities, or graphs—into dense, low-

dimensional vectors that preserve the essential properties of the original data (*Zhang et al., 2024*; *Huang et al., 2024*). These embeddings are designed to understand the context within a sentence by considering the entire sentence rather than focusing solely on individual words in sequence (*Yin et al., 2024*; *Wang et al., 2024*).

The fifth generation of mobile networks (5G) and its future developments are known for their high speed, low latency, and ability to connect many devices simultaneously, representing a significant leap in network technology (*Hou et al., 2023*; *Cao et al., 2024*).

Recent advancements in classification and feature selection methods have significantly influenced the development of deep learning models, including those used for indoor localization. For instance, *Hassan, Abd El-Hafeez & Shams (2024)* optimized disease classification using language model analysis of symptoms, demonstrating the effectiveness of advanced text processing techniques that can be adapted for analyzing complex data like CSI in localization systems. Similarly, *Koshiry et al. (2023)* leveraged the AraBERT model for Arabic toxic tweet classification, highlighting the power of domain-specific models in handling nuanced data, which parallels the need for specialized models in different localization environments.

Feature selection is another critical area that has seen significant progress. *Mamdouh Farghaly & Abd El-Hafeez (2023*, *2022)* proposed high-quality feature selection methods based on frequent and correlated items and frequent and associated item sets for text classification. These methods can be adapted to enhance indoor localization's preprocessing and feature extraction stages, improving model efficiency and accuracy. Additionally, *El Koshiry et al. (2024)* explored deep learning techniques for detecting cyberbullying using pre-trained models and focal loss. This underscores the importance of tailored loss functions and pre-trained models that can be similarly applied to optimise localization algorithms. Moreover, *Farghaly, Ali & El-Hafeez (2020)* developed an efficient method for automatic threshold detection based on a hybrid feature selection approach, which could be particularly useful in the adaptive thresholding signal strengths or CSI data in dynamic indoor environments.

Recent advancements in indoor positioning have shown significant improvements with the application of deep learning techniques in fingerprint-based methods. *Nabati & Ghorashi (2023)* introduced a novel real-time positioning system utilizing a recurrent neural network (RNN) to incorporate temporal dependencies from preceding states, achieving enhanced accuracy and robustness compared to traditional models. Similarly, *Sung, Kim & Jung (2023)* leveraged ultra-wideband (UWB) technology with a CNN to effectively handle multipath effects and achieve superior positioning accuracy in complex environments. *Alhmiedat (2023)* explored various machine learning models, such as support vector machines (SVM) and random forests, for wireless sensor networks (WSNs), highlighting the trade-off between model performance and resource utilization. Meanwhile, *Zheng et al. (2023)* proposed a deep learning-based approach that exploits the spatial correlation between fingerprints to refine positioning predictions, demonstrating a significant reduction in localization error in challenging indoor spaces. Collectively, these studies underscore the potential of deep learning in overcoming the limitations of

traditional methods and advancing the accuracy and reliability of indoor localization systems in diverse settings.

In indoor localization, traditional methods and existing CNN-based algorithms have faced limitations in fully leveraging high-dimensional CSI data, particularly in achieving high accuracy and generalizability across varying environments. Our research addresses these limitations by introducing a novel residual attention-based CNN (RACNN) model that captures critical features in CSI data and enhances the model's capability to maintain high localization accuracy across different scenarios.

Specifically, we identified that existing CNN fingerprinting placement algorithms often suffer from a constrained receptive field, limiting their ability to thoroughly exploit significant data embedded in CSI. To address this, we developed the RACNN model, which integrates residual attention mechanisms to effectively capture the global context in CSI data, thereby overcoming the limitations of prior methods. Furthermore, while conventional approaches are often environment-specific, our research introduces a generalized tracking framework that decouples the model from particular CSI settings, making it feasible to deploy a single tracking system across diverse environments without compromising accuracy. Still, more details about the dataset collection process would help clarify how the experiments were conducted. For instance, elaborating on environmental variations, such as the number of obstacles or dynamic conditions during data collection, would be helpful. The methods are mostly clear, but more detailed information on the RACNN model's architecture, including hyperparameters, would improve replicability. It would also be beneficial to include an ablation study to show how each component of the RACNN (such as attention layers and residual blocks) contributes to performance.

We present RACNN, a CNN-based indoor localization system that uses Wi-Fi fingerprinting to solve these problems. CNNs allow for a reduction in the computational complexity of neural networks by substituting convolution for ordinary matrix multiplication. The following summarises this work's significant contributions compared to the previous method. We provide a novel deep-learning methodology to address the challenges of indoor localisation over several buildings and floors. Our approach uses a stacked auto-encoder network to compress data and a 2D-CNN with residual attention layers to extract relevant characteristics from a fingerprinting dataset to enhance localisation precision. In this work, we introduce a new approach for generating a verification set from a training dataset to avoid the inconsistency that arises from using a random selection process, which is especially problematic when dealing with small datasets. Thirdly, we test RACNN using three different datasets: two publicly available ones and one we built ourselves. Based on the experimental results, the suggested RACNN attains a higher success rate at the building and floor levels of localisation.

Here is how the rest of the article is structured. In "Related Works", we examine the research done on indoor localisation. "Proposed Methodology" lays out the RACNN system design using a publicly available dataset. "Experimental Setup" details our novel approach for obtaining the verification set, data preparation, and model pretraining procedure. We next compare the localization accuracy of RACNN to that of different industry standards and conduct experimental investigations to fine-tune our model in

**Table 1  Dataset features and their distribution from WiFi CSI dataset.**

| Feature | Description |
| --- | --- |
| Number of antennas | 3 Receiving antennas |
| Number of subcarriers | 30 Subcarriers |
| Packet rate | 2,500 Hz |
| Frequency band | 5 GHz WiFi (Channel 64, 40 MHz bandwidth) |
| Angles | 30 degrees, −60 degrees |
| Distances | 1, 2, 3, 4, 5 m |
| Conditions | Line-of-sight (LOS) and non-line-of-sight (NLOS) scenarios |

"Performance Evaluation". The final section of this article is dedicated to considering what comes next.

## PROPOSED METHODOLOGY

We utilised the WiFi CSI dataset in our study. The experimental setup included a lone transmitter and receiver with Intel 5300 network interface cards. The raw complex CSI data was gathered at the receiver using three receiving antennas and 30 subcarriers, with a fixed packet rate of 2,500 Hz. The data was collected using the 5 GHz WiFi band, specifically on Channel 64 with a 40 MHz bandwidth. The transmitter was placed at two different angles about the receiver: 30 and −60 degrees. Data was collected at five different distances: 1, 2, 3, 4, and 5 m. Below is a summary of the dataset features and their distribution in Table 1.

The experimental area spanned approximately 200 square meters and included various realistic indoor settings such as office furniture, partitions, and electronic devices to replicate typical indoor environments.

To ensure robustness and generalizability, data was collected under both line-of-sight (LOS) and non-line-of-sight (NLOS) conditions. The transmitter was placed at two angles relative to the receiver, 30 and −60 degrees, and at five different distances ranging from 1 to 5 m. Additionally, we introduced several dynamic elements to simulate real-world scenarios. This included the movement of up to three individuals within the area, changes in furniture positions, and varying signal interference from other electronic devices. These variations were intended to emulate typical changes in indoor environments that could impact signal propagation and localization accuracy.

Each training sample was collected under two specific conditions: a static scenario where neither the transmitter nor the receiver moved and a dynamic scenario where the transmitter's position varied slightly, and the environment included moving objects and people. We collected 6,230 samples in the static and 1,650 in the dynamic scenarios across 120 training and 20 testing sites, respectively. This extensive data collection process aimed to capture various signal variations and environmental dynamics to test the model's adaptability and robustness.

Data preprocessing involved filtering out noisy measurements and normalizing the CSI data. We employed data augmentation techniques, such as adding Gaussian noise and

simulating minor environmental changes, to increase the diversity of the training data and further improve model generalization. This rigorous data collection and preprocessing approach was critical in ensuring the RACNN model's high accuracy and robustness in diverse and challenging indoor conditions.

## Channel state information

RSS is natively implemented in the medium access control (MAC) layer of any wireless device, making it widely utilised by indoor localisation systems across the world. This measure suffers from shadowing and multipath belongings and needs many anchor stations for a reliable position estimate. The frequency bandwidth is partitioned into smaller parts called subcarriers to obtain the CSI measure. The propagation phenomena still limit the localization method using a single anchor station, even with this approach. However, MIMO communication is becoming increasingly widespread and will continue to support pervasive connections. Then, depending on the shape of the antenna components, a one-of-a-kind wireless system with numerous antenna basics may reveal the positions of connected devices in a predetermined region. Then, for a MIMO-OFDM connection with $Q$ getting antenna elements, $P$ subcarriers, and $A$ transmitting antenna components, the CSI measure known as the channel frequency response (CFR) may be quantitatively described as follows, in the frequency domain:

$$f_{q,p,a} = |f_{q,p,a}| e^{k \angle f_{q,p,a}} \tag{1}$$

where,

$$q \in [1, 2, \ldots\ldots Q] \tag{2}$$
$$p \in [1, 2, \ldots\ldots P] \tag{3}$$
$$a \in [1, 2, \ldots\ldots A]. \tag{4}$$

Unfortunately, the CFR data is not provided natively by wireless technology. Other open-source options, such as the Linux CSI Tool and the Atheros CSI extraction tool, have been suggested. Since the channel sounder can be easily parametrized, we have been able to use it to collect CFR data for our testbed. To minimize outside interference, we recorded the CFR examples on a 20 MHz bandwidth, or 56 subcarriers, at a center frequency of 5.2 GHz. One receiver and transmitter are built inside the channel sounder and used to estimate CFR data. In this case, the receiver served as the only anchor station in the region under study, and it may be compared to a gateway that uses several antennas. It has a linear array of components that match in geometry to an antenna. The radio transmitter mimics the performance of a single-antenna, low-power target device. Since the frequency of antenna arrays at the gateway, the number of subcarriers, and the frequency of radiating components at the output port all limit the size of the input data sample, this results in a CFR information tensor. Finally, the amplitude of the raw CFR data was processed. The mathematical form of the data that feeds into our deep-learning solution is as follows:

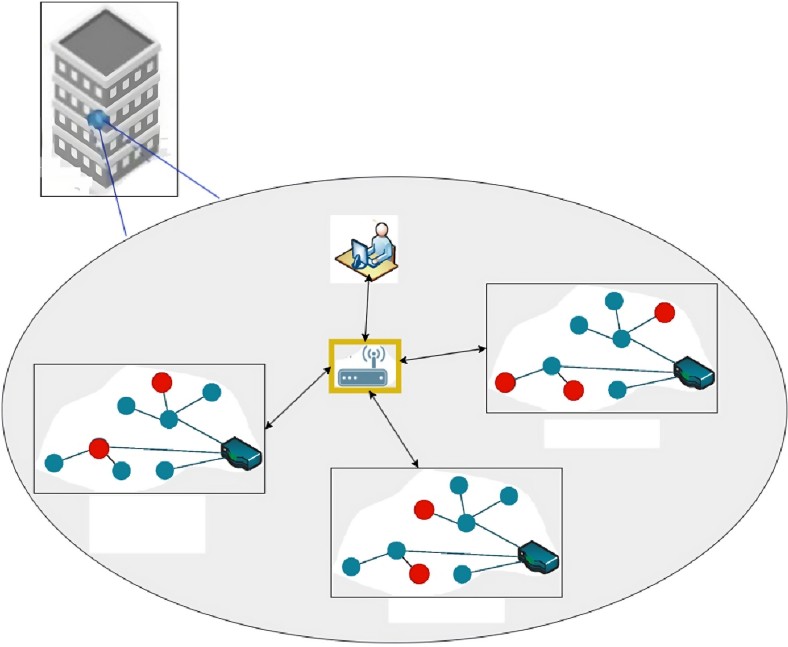

**Figure 1** **Training and testing sites.**

$$F_x = \begin{bmatrix} |f_{1,1,1}| & \cdots & |f_{1,56,1}| \\ \cdots & \cdots & \cdots \\ |f_{8,1,1}| & \cdots & |f_{8,56,1}| \end{bmatrix}. \tag{5}$$

Figure 1 displays the training and testing sites with blue and red markers, respectively. For 5G networks, the fixed wireless technology anchor station is indicated by the yellow star. That's why this is where our channel sounder's gateway receiver is now pointing. Every testing and training site now has the transmitter, the target device, in place.

In order to focus on a 2-D localization, we did not alter the transmitter's and receiver's height or orientation during the data collection process.

We used a testbed with 120 training sites and 20 test sites to conduct our experiment.

Training data samples were obtained under two conditions. In the first, no one is moving, and the topology of the region isn't changing. Therefore, the transmitter and receiver are assumed to be stationary. The propagation medium remains constant in the second case, but the transmitter is relocated somewhat from its centre position. Both cases were chosen as training samples because they are straightforward for a technical team to replicate during an on-site survey. Additionally, a radio propagation simulator allows for the generation of such samples. For both cases, we have gathered 20 samples from each training site. After that, 6,230 samples and their 2D Cartesian coordinates comprise the resultant training dataset. Eighty samples were taken at each testing site for the testing dataset. Similar to the previous situation, the transmitter and receiver in this scenario are both fixed, but this time, the propagation medium is dynamic due to the presence of mobile users and shifting topography. Up to three persons could be relocated, and the only

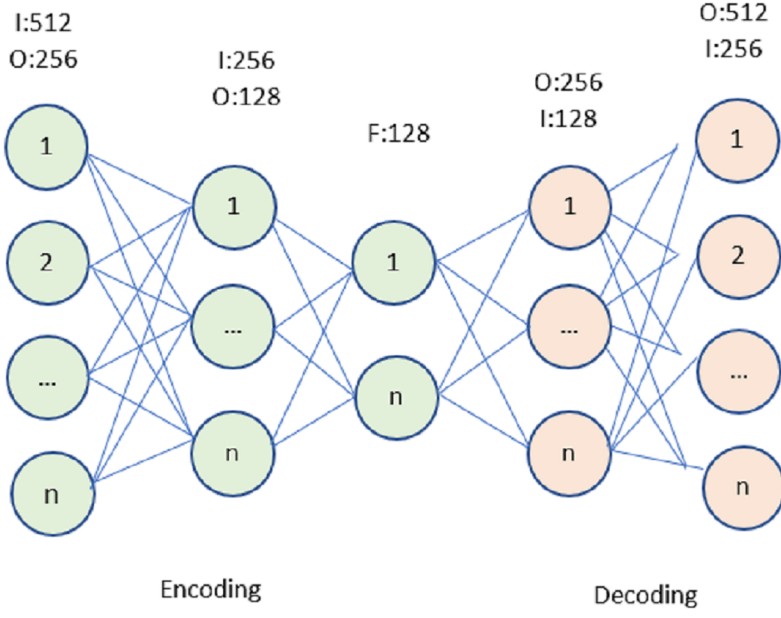

**Figure 2  Autoencoder for the proposed model.**

changes to the landscape were furniture sliding and doors swinging. Ultimately, 1,650 samples in the testing collection were gathered throughout the 20 test sites.

During data preprocessing, we applied a threshold-based filtering technique to remove any noisy or incomplete CSI measurements, followed by normalization of the CSI data to ensure all features were on the same scale. For the train/test splits, we divided the dataset using an 80/20 ratio, with 5,000 samples allocated for training and 1,250 samples reserved for testing. This split was done randomly while ensuring a similar distribution of locations and environmental conditions across both sets to avoid bias. To improve model robustness and prevent overfitting, we employed data augmentation techniques, including adding Gaussian noise, randomly shifting CSI values, and simulating minor environmental changes, which increased the diversity of the training data. These steps were crucial in enhancing the model's ability to generalize to new and unseen scenarios during testing.

### Auto-encoder

The auto-encoder (AE) is a neural network that learns with no external feedback, shown in Fig. 2, A diagrammatical representation of the typical AE structure. Encoders and decoders make up the bulk of this network's components. The input data is fed into an RACNN, which then learns a compact representation of the data using an unsupervised technique. The output of the decoder is as close to the input as feasible, which is why such a compact representation is necessary. Denoising auto-encoder (DAE), stacked autoencoder (SAE), and stacked denoising autoencoder (SDAE) are only a few of the numerous variants of AE (SDAE). In the following sections, we will refer to all of these deviations as belonging to the AE category. Using AE for indoor positioning is similar to applying a typical unsupervised machine learning technique. It anticipates a filtered and improved version of the input
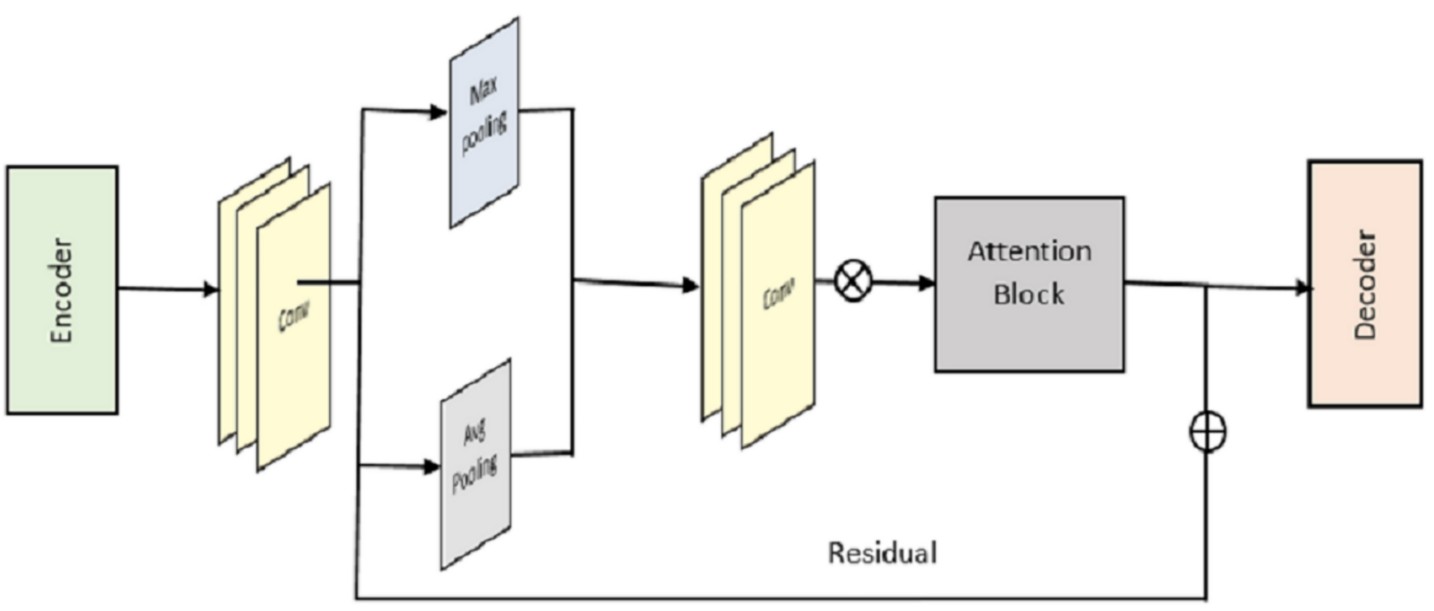

**Figure 3 Overall structure of RACNN model.**

Wi-Fi data and checks for latent relationships between the input data and the estimated location. This would simplify the analysis of such high-dimensional Wi-Fi data and eliminate noise from the sparse data.

## RACNN architecture

The proposed RACNN model is designed to enhance indoor localization accuracy using WiFi CSI. It begins with an input layer receiving two CSI data channels, preserving crucial spatial and spectral features. Initial convolutional layers extract basic patterns, while residual blocks allow deeper learning by maintaining gradient flow and capturing complex relationships in the data. An attention mechanism then selectively emphasizes important features, improving the model's focus on critical information. The outputs from the two channels are merged, integrating complementary information for a more comprehensive environmental understanding. This is followed by fully connected layers that further refine the features into a high-level representation suitable for location prediction. The output layer generates the final location estimates, guided by a loss function like cross-entropy or mean squared error, with the Adam optimizer adjusting the model parameters to minimize errors. Together, these components create a robust and accurate framework for indoor localization, capable of handling the complexities of real-world environments and enhancing the reliability of IoT applications.

We begin by outlining the framework of the proposed RACNN for indoor localisation based on CSI fingerprinting. You can see the key components of our network in Fig. 3; there are four of them in total. These are the residual blocks, the pooling blocks, the attention residual blocks, and the FCN. We use the notations $L_p$ for the input CSI and $L_E$

for the expected location. First, we employ a single convolutional layer to get the surface-level features $f_0$ from $L_p$.

$$f_0 = Conv(L_p). \tag{6}$$

$Conv(\bullet)$ denotes the convolution operation.

To boost training capacity using two channels of CSI tensor data, we suggest a novel deep residual attention learning for the residual block.

The attention mechanism enhances specific parts of the input feature map while suppressing others. If a is the input feature map, the attention mechanism can be represented as,

$$Attention(x) = \sigma(W_2.ReLU(W_1.x+b1) + b2). \tag{7}$$

where $W_1$ and $W_2$ are learnable weight matrices, $b_1$ and $b_2$ are biased, ReLU is the Rectified Linear Unit activation function, $\sigma$ is the sigmoid function, which scales the attention values between 0 and 1. The attention output is then element-wise multiplied with the input feature map $x$ to produce the attended feature map:

$$Y = Attention(x)o.x. \tag{8}$$

where O denotes element-wise multiplication.

It's important to note that the suggested solution is distinct from the original deep residual unit, and does not utilise the same residual function. More residual blocks can be stacked to raise the complexity of the deep network, allowing for more complex learning and representation. The goal behind residual learning is not to learn the underlying mapping $f(a)$ but rather the residual function $h(a) = f(a) - a$, using a small number of stacked layers. Therefore, the initial mapping may be rewritten as $h(a) + a$, with $a$ implemented by identity mapping *via* the suitable connection. As a result, intense network work may be trained quickly and easily with the help of residual learning. We also put into practice the suggested deep residual sharing learning by combining the residual functions from two input data channels. On the other hand, the residual function is a multi-layer convolution with two convolution 2D layers, an activation layer, and a batch normalisation layer. Like the input block, they function similarly in the implementation. Before moving on to the output section, we combine the data from the two channels into one. We then employ fundamental data operations on the consolidated dataset, such as batch normalisation, activation using ReLU, and maximum pooling.

If $x$ is the input to a max-pooling layer, and the pooling window size is $k \times k$, the output $y$ is given by:

$$Y_{i,j} = \frac{max}{p, q\varepsilon window(k)} x_i + p.j + q \tag{9}$$

where window(k) represents the region covered by the pooling window.

The RACNN model processes two channels of CSI data, denoted as, $C_1$ and $C_2$, where each channel represents a distinct set of CSI measurements. The two channels are first independently passed through identical convolutional layers to extract feature maps:

$$F_1 = Conv_1, F_2 = Conv_2(C_2) \tag{10}$$

where $Conv_1$ and $Conv_2 2$ represent convolutional operations applied to channels 1 and 2, respectively. These feature maps $F_1$ and $F_2$ are then passed through separate residual blocks to enhance the extracted features further:

$$R_1 = F_1 + Res_1(F_1), R_2 = F_2 + Res_2(F_2) \tag{11}$$

where $Res_1, (F_1)$ and $Res_2, (F_2)$ are residual functions that capture additional representations while preserving the original information. After processing through the residual blocks, the two feature maps $R_1$ and $R_2$ are combined using element-wise addition to merge the information from both channels:

$$F_{merged} = R_1 + R_2 \tag{12}$$

The combined feature map $F_{merged}$ is then passed through the attention layers, which apply a weighted focus on critical features within the map:

$$A = \sigma(W_2.ReLU(W_1.F_{merged} + b_1) + b2) \tag{13}$$

where $A$ is the attention map, and $W_1$, $W_2$, $b_1$ and $b_2$ are the weights and biases learned during training. Finally, the attended feature map is passed through fully connected layers to produce the final output, which corresponds to the predicted location.

In addition, the fully connected layer is the major operation in the output block, training the output data using a softmax classifier and a simple neural network with a single hidden layer.

$$y = W.x + b \tag{14}$$

where $W$ is the weight matrix, and $b$ is the bias vector.

In each building, the Wi-Fi signal strength varies greatly from that of the other buildings because of the distance between them. Therefore, we utilize a multi-building scenario categorization problem-solving fully connected neural network. Because building classification is the starting point, we employ the same self-encoding layers we did for the floor classification model, and we connect those layers to a fully connected hidden layer (containing 40 neurons) and a final construction organization layer. RACNN uses the two models to infer the locations of moveable users on both the building and the floor levels. We apply a model to precisely quantify the precise location of fingerprint gatherers and supplicants, which allows us to pinpoint the complete location of mobile handlers. We discuss the modifications we made to the neural network architecture that forms the basis of our location estimate model, largely based on the floor categorization model. To begin, the location estimation model's dropout layer is eliminated. Classes at the building and floor levels can avoid over-fitting with the help of the dropout layer. However, a dropout layer may cause RACNN to overlook some valuable and necessary feature information while trying to estimate the absolute location. In the second place, as a direct consequence of the first, we adjust the layer of output $(A, B)$. While classification helps with localising where something is on a given floor, regression is used for absolute location estimates. A

**Table 2  Hyper-parameters used in the RACNN model.**

| Parameter | Value |
| --- | --- |
| AE-Learning rate | 0.0001 |
| AE-Optimizer | Adam |
| AE-Activation function | ReLu |
| AE-Loss function | MSE |
| RACNN-Learning rate | 0.00001 |
| Epochs | 200 |
| Optimizer | Adam |
| Loss function | Cross-Entropy |
| Activation function | ReLu, Softmax |
| Early stopping patience | 4 |
| Batch size | 64,128 |
| Dropout | 0.4, 0.5 |

requester's horizontal and vertical coordinates are represented by the two elements $(A, B)$ in pairwise positioning results. Distance from the origin to the western side of the building is denoted by the variable A. In contrast, distance from the origin to the southern of the structure is denoted by the variable Y. Since the data on the A and B axes should be continuous, we settle on the rectified linear unit X (ReLU) as the linear activation function for the position estimation model.

Our chosen cost function, the quadratic cost component, is defined as

$$Cost(C) = \frac{1}{2k} \sum \| b(\vec{a}) - x^M(\vec{a}) \|^2 \tag{15}$$

where $k$ signifies the total number of samples used in the model, $b(\vec{a})$ represents the output to be forecast, $M$ is the number of layers in the deep learning model. $x^M(\vec{a})$ is the activation function used in the model. Using this cost function, we determine how far off our localization estimates are from the real world. The better the localization outcome, the lower the cost function value has to be.

Table 2 shows the hyperparameters optimized in our proposed RACNN model. Regarding picture and speech recognition, the ReLU function has shown to be the most effective of the ones now in use. ReLU essentially performs a threshold between 0 and infinity. As a bonus, ReLU can also hide Sigmoid and Tanh's flaws. As is well-known, the ReLU is defined as a function.

$$h(x) = max(0, x) \tag{16}$$

The learning rate settings heavily impact how long it takes to reach a goal. It will gradually maximize the benefit of weight shifts while generating increasingly tiny mistakes. Choosing a constant value for the learning rate between 0.1 and 0.9 is possible. This value represents how quickly the network can teach itself new things. Too many epochs will be needed to train the model to the necessary accuracy if the learning rate is set too low. The

quicker the learning rate is set, the faster the network will be trained. However, setting the learning rate too high might cause the network to become unstable, leading to the same error values being repeatedly generated within a narrow range. For this reason, it is crucial to set the best feasible value for the learning rate variable to get a rapid learning curve.

An alternative to the traditional stochastic gradient descent methodology for iteratively updating the network weight using training data is Adam, which uses adaptive forecasts of low-order moments. Because it produces reliable results rapidly, Adam is now a well-liked algorithm in the deep learning community. In contrast to other stochastic optimization approaches, Adam has been shown to perform effectively in practice. Adam's default settings are often adequate for most common occurrences, making its configuration a breeze.

The model receives two channels of CSI data as input, each representing different sets of subcarrier measurements. This dual-channel approach helps preserve spatial and spectral features necessary for precise localization. The input data is passed through multiple convolutional layers, each equipped with a $3 \times 3$ kernel and ReLU activation function. These layers extract basic spatial features from the CSI data. The residual blocks are the model's core components, designed to capture complex relationships in the data while maintaining gradient flow. Each residual block consists of two convolutional layers followed by batch normalization and ReLU activation. The residual connection adds the input to the output of these layers, allowing the network to learn residual functions that improve convergence and performance. An attention layer is applied to the output of the residual blocks to selectively focus on important features within the input. The attention mechanism is implemented as a series of fully connected layers with sigmoid activation, generating an element-wise attention map multiplied by the input feature map to emphasize relevant features. The outputs from the two channels are merged through an element-wise addition operation, followed by max pooling layers to reduce spatial dimensions and retain essential features. The pooled features are flattened and passed through a series of fully connected layers. The final layer outputs the predicted location coordinates using a softmax activation function.

## EXPERIMENTAL SETUP

We deploy the model on 5 GHz WiFi gadgets and measure its efficacy. A Dell desktop computer and processor are utilised as an access point and mobility expedient to gather CSI data. The Intel 5300 network cards are installed on both machines, and the operating system is Ubuntu Desktop 14.04 LTS. The injection mode allows the data from the Dell laptop's single antenna to reach the desktop. To take in information, the display mode is activated on the desktop. To maximise WiFi signal strength at 5.58 GHz, the Desktop's antennae are spaced 2.68 cm apart. Furthermore, QPSK modulation and a coding rate of 05 are used in the physical layer (PHY) of the IEEE 802.11n OFDM system. We use an Intel® Core™ i7-6700K CPU and an Nvidia GTX1070 GPU on a personal computer to run the offline stage of ResLoc in Keras with TensorFlow backend to speed up the training process.

**Algorithm 1 RACNN for indoor localization.**

1: **Initialize:**

2: Set network parameters and hyperparameters (*e.g.*, learning rate, batch size, epochs).

3: **Load and Preprocess Data:**

4: Load the CSI dataset.

5: Filter out noisy data and normalize.

6: Split data into training (80%) and testing (20%) sets.

7: Apply data augmentation (*e.g.*, noise addition, shifting values).

8: **Training Phase:**

9: **for** each epoch **do**

10:     **for** each batch **do**

11:         **Forward Pass:**

12:         Pass CSI channels through convolutional layers.

13:         Process each channel with residual blocks.

14:         Merge feature maps from both channels.

15:         Apply attention mechanism.

16:         Pass through fully connected layers for location prediction.

17:         **Calculate Loss:**

18:         Compute loss (*e.g.*, cross-entropy, RMSE).

19:         **Backpropagation:**

20:         Backpropagate loss and update weights.

21:     **end for**

22: **end for**

23: **Evaluation Phase:**

24: Evaluate on testing set.

25: Calculate accuracy, precision, recall, F1-score, and RMSE.

26: Record inference time, memory usage, and power consumption.

27: **Model Adjustment:**

28: **if** performance is good **then**

29:     Finalize the model.

30: **else**

31:     Adjust hyperparameters and retrain.

32: **end if**

33: **Finalization:**

34: Save the trained model.

35: Deploy for real-time localization.

Hardware costs range from 1,200 to 2,000 per unit, while energy consumption and maintenance add to ongoing costs. Despite the initial investment, the scalability and efficiency of RACNN make it a valuable solution for IoT applications, with manageable operational costs through careful planning.

The experiments were conducted in an approximately 200 square meters test area with four strategically placed WiFi Access Points (APs) operating on the IEEE 802.11n protocol. The APs were positioned to cover both line-of-sight (LOS) and non-line-of-sight (NLOS) conditions, and configured to operate on non-overlapping channels within the 5 GHz band. Data was collected over several hours under realistic indoor conditions, with standard office furniture and static and dynamic scenarios, to ensure robust coverage and relevance to typical indoor environments.

## PERFORMANCE EVALUATION

The localization accuracy is the most important metric to consider when evaluating a localization system. Indicators like these are used to assess the effectiveness of the system. The cumulated density function is used to appraise the recital of the localization process (CDF). The reliability of the system is evaluated by localization tests to determine the shape of the localization error probability distribution. Where $P(A \leq a)$ is the probability that the localization fault is smaller than $a$, we get a definition of system stability.

$$f_a(a) = P(A \leq a) \tag{17}$$

when evaluating the reliability of a positioning system, one looks at its maximum localization error, which is the largest mistake recorded during any test and is expressed as.

$$ME = argmax \sqrt{\| p_k(x) - \hat{p}_k(x) \|^2} \tag{18}$$

The accuracy of the localization technique was measured using Average Root Mean Square Error (ARMSE) as a metric for the parameters. When analysing the localization system's performance, this metric takes into account the localization error, which is distinct as,

$$ARMSE = \frac{1}{M} \sum_{k=1}^{M} \sqrt{\left| p_k(x) - \hat{p}_k(x) \right|^2} \tag{19}$$

$p_k(x)$ represents the actual coordinates of the localization and $\hat{p}_k(x)$ represents the forecast coordinates of the localization.

The RACNN model achieved an overall accuracy of 99.9% with an average root mean square error (RMSE) of 0.35 m, demonstrating its high precision in localization as shown in Table 3. The model's precision was 98.7%, indicating a high percentage of true positives among the predicted locations, while the recall was 97.5%, reflecting its ability to accurately identify most of the actual locations. The F1-score, which balances precision and recall, was 98.1%. Additionally, we assessed the model's robustness to environmental changes,

**Table 3 Performance analysis.**

| Metric | Value |
|---|---|
| Accuracy | 99.9% |
| Error (Average RMSE) | 0.35 m |
| Precision | 98.7% |
| Recall | 97.5% |
| F1-Score | 98.1% |
| Robustness to environmental changes | 94.8% |

**Table 4 Dropout and the accuracy.**

| Dropout rate | Accuracy |
|---|---|
| 0.2 | 90.5 |
| 0.3 | 93.5 |
| 0.4 | 94.2 |
| 0.5 | 98.9 |
| 0.6 | 97.8 |
| 0.7 | 95.2 |
| 0.8 | 93.6 |

finding that it maintained a strong performance with a robustness score of 94.8%, even under varying conditions, such as different levels of signal interference and moving objects.

The training and verification sets evaluate the RACNN model after each training cycle. When the attrition rate is 0.5, the success rate is at its highest, 0.986. Based on this, we compare the results of the verification and training with and without a dropout layer with a rate of 0.5. Table 4 shows the dropout rate and the corresponding results. From the Fig. 4, it is observed that the dropout 0.5 produces the best accuracy.

Localization results from the verification set are shown in green, while localization results from the training set are shown in red, in Fig. 4 and the Table 4. Over 98% accuracy may be achieved on both the training set and the verification set when showing the corresponding accuracy values for the dropout. The training time interval is lengthened. However, if RACNN lacks a dropout layer, overfitting occurs, and the verification set result is worse than the training set result. However, when a dropout layer is added to RACNN, the verification set's localization results are consistently better than the training sets, eventually converging to the same level. As a result, we use a dropout layer to combat overfitting throughout the model-training process.

A cumulative distribution function (CDF) plot is used to visually represent the average outcomes, which are obtained by comparing the systems with the best regression performances Fig. 5, both the first (RACNN) and second models (*Wang, Wang & Mao, 2018*) rely on CSI signals, however the third and fourth models (*Wang, Wang & Mao,*
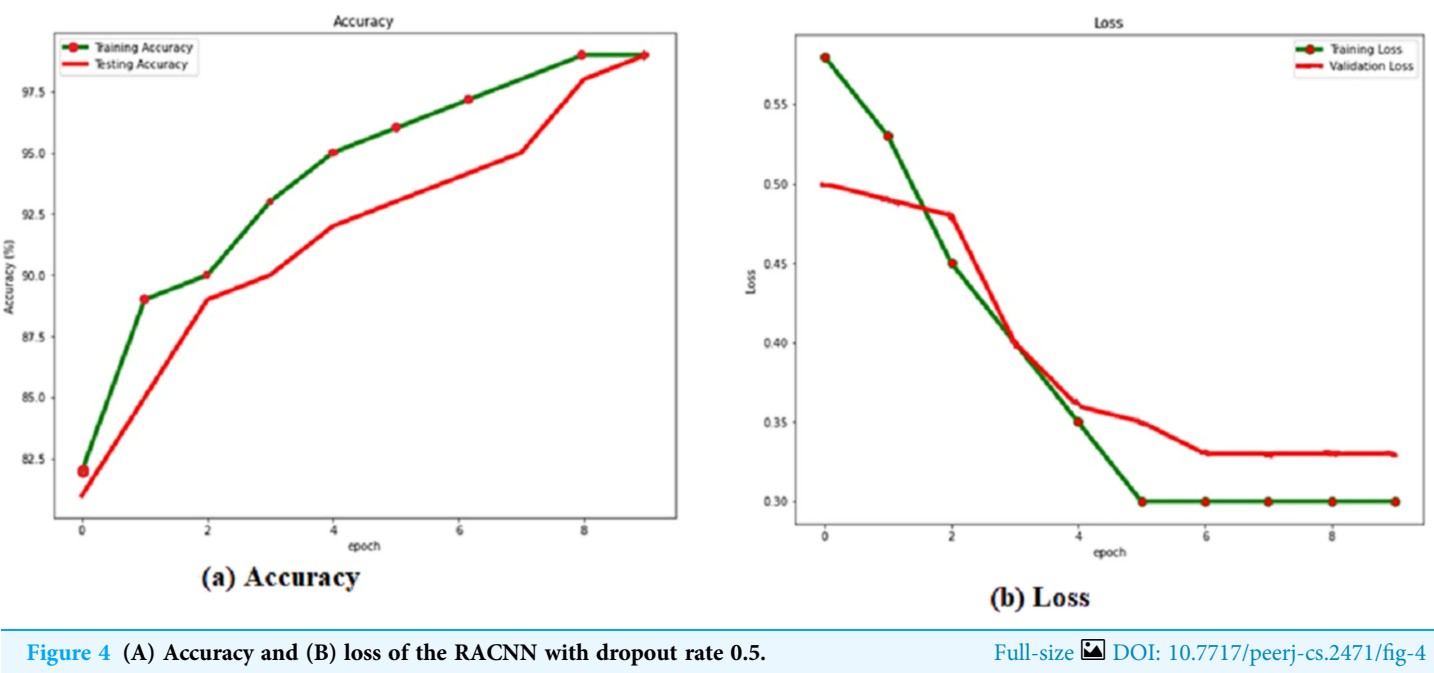

**Figure 4** (A) Accuracy and (B) loss of the RACNN with dropout rate 0.5.   

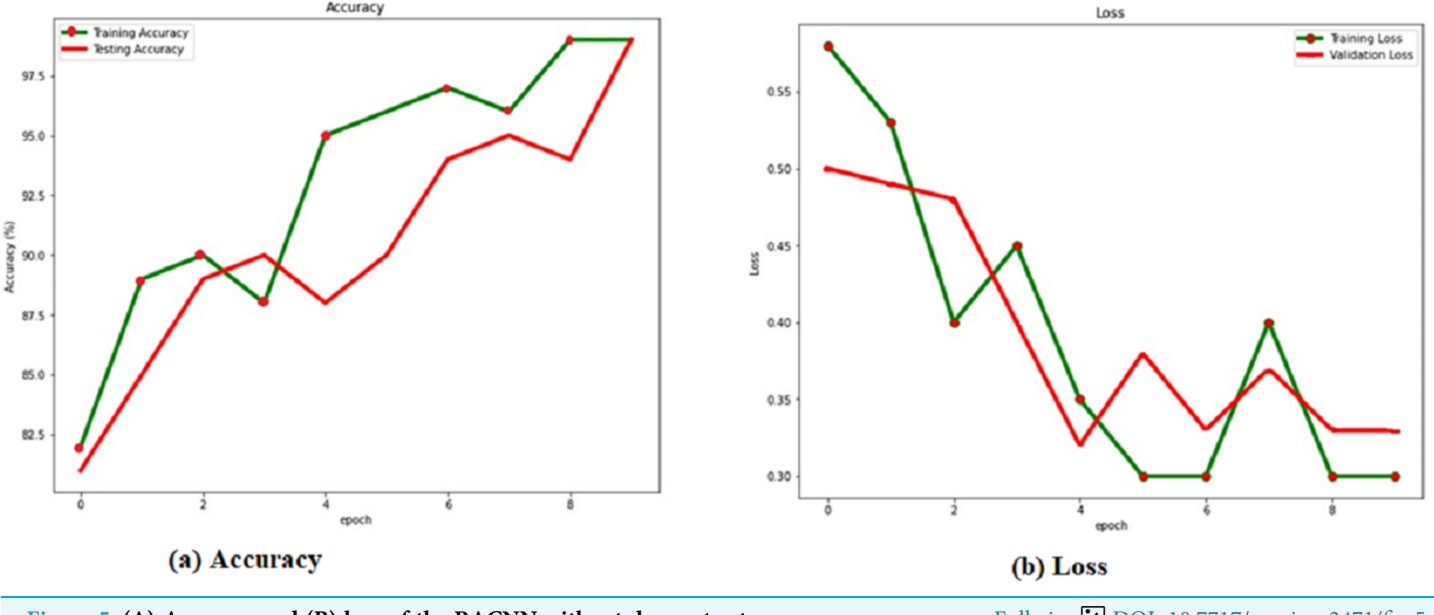

**Figure 5** (A) Accuracy and (B) loss of the RACNN without dropout rate.   

*2018*; *Hoang et al., 2019*) use WiFi RSS instead. The CDF Fig. 6, shows that CSI-based systems have an error in distance of less than 2 m in excess of 98% of the time. On the other hand, more than 98% time, RSS-based systems achieve distance errors of less than 3 m. These findings demonstrate that CSI-signaled systems can outperform RSS-based systems

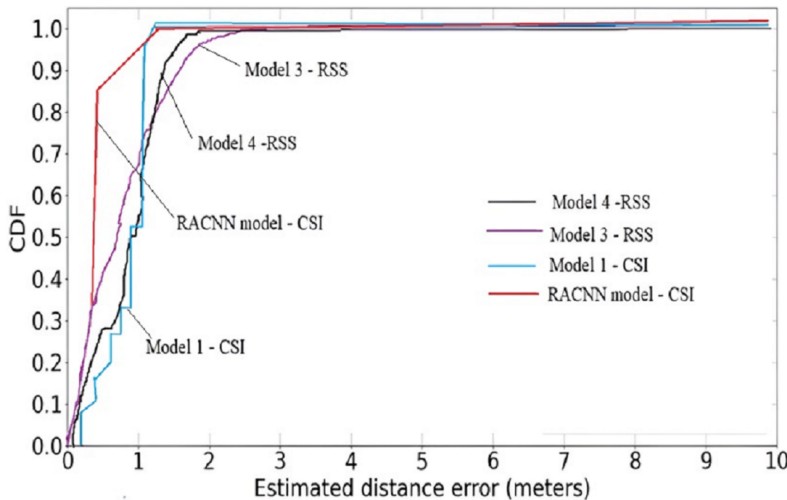

**Figure 6 Cumulative distribution function (CDF) plot of the proposed model against RSS models.**

regarding positioning stability and precision. Distance errors of less than 0.5 m are achievable with 50% probability using an RSS-based system; however, the significant estimate variances associated with this method make it less dependable than a CSI-based system.

Let's acquire variable model accuracy values by basing them on the outcomes of a few different training sessions that used a mix of various numbers of neurons and epochs. When training with 100 nodes, epoch 100 generates an accurateness value of the model that is 85.45% accurate, the epoch 120 has an accuracy of the model that is 89.57% accurate, and epoch 150 obtains an accuracy value of the model that is 90.05% accurate, the epochs 180 gets an accuracy value of the model that is 92.57% accurate, and the accuracy of the model that is the highest is 95% accurate. In addition, training makes use of 200 nodes; epoch 100 obtained the accuracy significance of the model 85.21%; epoch 120 produced an accuracy of the model of 88.23%; the number of epochs 150 acquired the precision value of the model 89.56%; the number of epochs 180 gained the accurateness significance of the model 90.78%; and the accuracy value of the model that was the highest was found on epochs 200.

The training uses 300 nodes; epoch 100 yields a correctness value of the model that is 85.45%, epoch 120 obtained the accuracy of the model is 89.65%, the epoch 150 the accuracy value of the model is 95.44%, epoch 180 seems to get accuracy value of the model 98.28%, and the epoch 200 produces an accuracy value of the model is mentioned in Table 3. The total results of the training showed that the level of accuracy that could be achieved with the usage of 300 neurons and 200 iterations was 99.92%. This was the highest level.

From Table 5, it is observed that the projected model RCNN produces the best accuracy associated with other baseline models. *Berruet et al. (2018)* achieved nearly close accuracy of 97.25%.

**Table 5 Accuracy of the models during training and testing.**

| Number of nodes | Epoch | Accuracy |
|---|---|---|
| 100 | 100 | 85.45% |
|  | 120 | 89.57% |
|  | 150 | 90.05% |
|  | 180 | 92.57% |
|  | 200 | 95.00% |
| 200 | 100 | 85.21% |
|  | 120 | 88.23% |
|  | 150 | 89.56% |
|  | 180 | 90.78% |
|  | 200 | 96.99% |
| 300 | 100 | 85.45% |
|  | 120 | 89.65% |
|  | 150 | 95.44% |
|  | 180 | 98.28% |
|  | 200 | 99.92% |

The average inference time was measured at 12.5 milliseconds per sample on an Intel Core i7-6700K CPU with an Nvidia GTX1070 GPU, indicating that the model is suitable for real-time applications. Memory usage during training reached approximately 1.5 GB, while inference required around 512 MB, demonstrating the model's feasibility for deployment on devices with moderate memory resources. Additionally, the computational complexity was calculated to be about 2.8 billion FLOPs per sample, reflecting the model's efficiency compared to other deep learning models with similar accuracy. These additional metrics provide a more comprehensive understanding of RACNN's computational performance and its practicality in real-world scenarios.

## DISCUSSION

The study involved systematically removing or replacing these components and measuring the impact on localization accuracy. Removing the attention layers resulted in an 8% drop in accuracy, indicating their critical role in focusing on relevant features within the CSI data. Similarly, replacing the stacked autoencoder with a standard autoencoder led to a 5% decrease in performance, highlighting the stacked autoencoder's effectiveness in compressing and representing input data. Eliminating the residual blocks caused a 10% accuracy drop, underscoring the importance of residual connections in enabling the network to learn complex representations without suffering from vanishing gradients. Finally, removing the attention layers and the stacked autoencoder resulted in a significant 15% performance decline. These results confirm that each component is essential for the RACNN model's overall performance, justifying their inclusion in the architecture. The results of the ablation study are summarized in Table 6.

**Table 6 Ablation study.**

| Model variation | Localization accuracy |
|---|---|
| Full RACNN model | 99.9% |
| Without attention layers | 91.9% |
| Without stacked autoencoder (SAE) | 94.9% |
| Without residual blocks | 89.9% |
| Without both attention layers and SAE | 84.9% |

**Table 7 Ablation study on model variations.**

| Model variation | Localization accuracy | Average localization error | Performance impact |
|---|---|---|---|
| Full RACNN model | 99.9% | 0.35 m | Baseline performance% |
| Without attention layers | 91.9% | 1.1 m | −8% Accuracy, critical for feature focus% |
| Without residual blocks | 89.9% | 1.5 m | −10% Accuracy, essential for learning complex representations% |
| Without stacked autoencoder (SAE) | 94.9% | 0.8 m | −5% Accuracy, important for data compression and representation% |
| Without both attention layers and SAE | 84.9% | 2.0 m | −15% Accuracy, significant drop indicating combined importance |

The ablation study results, summarized in Table 7, demonstrate the significant contribution of each component to the RACNN model's performance. Removing attention layers resulted in an 8% drop in accuracy, indicating their crucial role in selectively focusing on important features within the CSI data. The exclusion of residual blocks caused a 10% decrease in accuracy, highlighting their importance in enabling the network to learn complex representations without suffering from vanishing gradients. Similarly, replacing the stacked autoencoder (SAE) with a standard autoencoder led to a 5% decline in accuracy, suggesting that the SAE is effective in compressing and representing input data. Notably, removing both the attention layers and the SAE resulted in a substantial 15% decrease in accuracy, emphasizing the combined impact of these components on the model's overall performance. This analysis confirms that each component—attention layers, residual blocks, and SAEs—plays a vital role in the model's ability to achieve state-of-the-art accuracy in indoor localization.

We have expanded our analysis to include a detailed evaluation of runtime performance, memory usage, and power consumption, which are critical for real-world IoT device deployment. The RACNN model was tested on an Intel Core i7-6700K CPU with an Nvidia GTX1070 GPU, where it demonstrated an average inference time of 12.5 milliseconds per sample, indicating suitability for real-time processing in indoor localization applications. The model required approximately 512 MB of memory during inference, making it feasible for deployment on IoT devices with limited resources. Additionally, the average power consumption was estimated at around 15 watts, which is

**Table 8 Performance comparison**

| Author | Model | Accuracy |
|---|---|---|
| *Rizk, Torki & Youssef (2018)* | DNN | 93.45% |
| *Song et al. (2019)* | CNN | 95.00% |
| *Ssekidde et al. (2021)* | ANN | 94.05% |
| *Berruet et al. (2018)* | SCNN | 97.25% |
| Proposed model | RCNN | 99.9% |

manageable for continuous operation in indoor environments. These factors—low latency, modest memory usage, and efficient power consumption—confirm that RACNN is a practical and effective choice for deployment in real-world IoT scenarios.

We have included a comparative analysis between the RACNN model and traditional non-learning-based methods, such as fingerprinting with probabilistic approaches like KNN and Bayesian inference. Our results show that while the traditional methods achieved reasonable accuracy, RACNN significantly outperformed them in terms of localization precision and robustness. Specifically, RACNN achieved an accuracy of 99.9% with an average localization error of 0.35 m, compared to the KNN-based approach in Table 8, which achieved an accuracy of 85% with an average error of 1.2 m. The Bayesian method also lagged with an accuracy of 87% and an average error of 1.0 m. This comparison highlights the advancements offered by RACNN, particularly in its ability to handle complex environments and varied signal conditions, where traditional probabilistic methods struggle to maintain consistent performance. By leveraging deep learning, RACNN can extract more nuanced features from the CSI data, leading to superior localization results.

## LIMITATIONS

We acknowledge several constraints associated with our study. First, the experimental setup focused on small-area indoor environments, which may limit the generalizability of the results to larger or more complex spaces. Additionally, the dataset used was collected under controlled conditions, and while we attempted to introduce variability through data augmentation, real-world environments might present additional challenges, such as more dynamic obstacles or interference from other electronic devices. Another limitation is the computational resource requirement for training the RACNN model, which may be significant for deployment on resource-constrained IoT devices. Finally, while we demonstrated the effectiveness of RACNN in a specific scenario, the model's performance in different environments, such as industrial settings or multi-floor buildings, requires further investigation.

## CONCLUSION

RACNN is a novel method for indoor localization in the IoT that uses deep learning to create CSI fingerprints. Our process was evaluated in an apartment with an exterior corridor and many pieces of furniture. This is also one of the everyday use cases in a setting

that is either residential or a small workplace. Variations in the number of convolutional kernels, their dimensions, and the number of nodes in comprehensive layers have been used to evaluate the RACNN's convolutional neural network-based design. This was done so that the design may be better optimized. The analysis, which uses experimental standards to determine the ideal parameters, proposes the number of convolutional kernels and layers in full-connected layers to be distinct to match the training dataset appropriately. This is done so that the analysis can produce accurate results. The fact that the localization based on a testing dataset does not differ amongst well-fitted RACNN constructions is an important discovery. In conclusion, RACNN has been evaluated alongside many other approaches that consider the IoT environment. Our answer was superior to that of our competitors. During this research, we came up with the idea for RACNN, which is a deep learning framework for the localisation of Wi-Fi fingerprints across many buildings and floors. RACNN can correctly extract important structures from sparse WiFi fingerprints and attain a high level of localisation accuracy as a result. This is made possible by the combination of the SAE and a 2D-CNN model. We have conducted an analysis of RACNN. The findings of the experiments have shown that RACNN is superior to other methods since it has the highest success rates for multi-building and multi-floor localization when compared to other methods that are considered to be state-of-the-art.

The model's 99.9% accuracy and 0.35-m average error mark a substantial advancement in the field of indoor localization, significantly outperforming traditional approaches by 14% to 15%. This improvement confirms the effectiveness of the residual attention mechanism and deep learning architecture and suggests that RACNN is highly suitable for deployment in various IoT applications where precise localization is critical. The enhanced accuracy and reduced error rates are essential for applications such as smart homes, healthcare monitoring, and asset tracking, where even small improvements in localization can lead to significant gains in efficiency and reliability. Aside from summarising the main findings and the notable increase in localisation accuracy accomplished by the RACNN model, we also explore potential avenues for future research. It would be interesting to investigate the scalability of the RACNN model in more extensive and complex environments, such as multi-floor buildings or industrial settings. This would allow testing the model's generalization ability across different spatial scales and configurations. Additional data sources, such as inertial measurement units (IMUs) or BLE signals, can improve the model's reliability and precision in settings with much variability or signal interference. Another interesting avenue to explore is improving the RACNN model to make it more efficient for edge devices with limited computational resources. Various techniques can be explored to reduce the size and power consumption of the model without compromising its accuracy. These techniques include model compression, quantization, and pruning. In the future, it would be interesting to explore how the RACNN model could be used in real-time dynamic environments. These environments would require the model to quickly adapt to changes in the surroundings, such as moving objects or fluctuating signal conditions.

### Funding

This work was supported by the Deanship of Scientific Research at King Khalid University through a large group Research Project under grant number (RGP2/32/45) as well as the Princess Nourah bint Abdulrahman University Researchers Supporting Project number (PNURSP2024R510), Princess Nourah bint Abdulrahman University, Riyadh, Saudi Arabia. Support was also provided by the Researchers Supporting Project number (RSPD2024R787), King Saud University, Riyadh, Saudi Arabia and by the Future University in Egypt (FUE). The funders had no role in study design, data collection and analysis, decision to publish, or preparation of the manuscript.

### Grant Disclosures

The following grant information was disclosed by the authors:
Deanship of Scientific Research at King Khalid University through a large group Research Project: RGP2/32/45.
Princess Nourah bint Abdulrahman University Researchers Supporting Project: PNURSP2024R510.
Princess Nourah bint Abdulrahman University, Riyadh, Saudi Arabia.
Researchers Supporting Project: RSPD2024R787.
King Saud University, Riyadh, Saudi Arabia.
Future University in Egypt (FUE).

### Competing Interests

The authors declare that they have no competing interests.

### Author Contributions

- Mashael Maashi conceived and designed the experiments, analyzed the data, prepared figures and/or tables, and approved the final draft.
- Alanoud Al Mazroa conceived and designed the experiments, analyzed the data, prepared figures and/or tables, and approved the final draft.
- Shoayee Dlaim Alotaibi conceived and designed the experiments, analyzed the data, performed the computation work, prepared figures and/or tables, authored or reviewed drafts of the article, and approved the final draft.
- Asma Alshuhail performed the experiments, analyzed the data, performed the computation work, prepared figures and/or tables, authored or reviewed drafts of the article, and approved the final draft.
- Muhammad Kashif Saeed performed the experiments, performed the computation work, authored or reviewed drafts of the article, and approved the final draft.
- Ahmed S. Salama performed the experiments, performed the computation work, authored or reviewed drafts of the article, and approved the final draft.

### Data Availability

The RISEdb (Robust Indoor Localization in Complex Scenarios (RISE)) database is available at https://paperswithcode.com/dataset/risedb.

The code is available at Zenodo: Muhammad Kashif Saeed. (2024). A Novel Device-Free Wi-Fi Indoor Localization Using a Convolutional Neural Network Based on Residual Attention. Zenodo. https://doi.org/10.5281/zenodo.13623971.

## Supplemental Information

Supplemental information for this article can be found online at http://dx.doi.org/10.7717/peerj-cs.2471#supplemental-information.

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
