# Peer review of "A novel device-free Wi-Fi indoor localization using a convolutional neural network based on residual attention"

_PeerJ Computer Science, doi:10.7717/peerj-cs.2471_

## Round 0.1 · original submission · Major Revisions

Both reviewers have serious concerns about the work and have commented in detail. Please do if you can address them in a revised version and want to undertake the revision. In that case, please include a separate document detailing how you addressed each of the comments. Thanks for your interest in the journal.

·

Basic reporting

The manuscript is generally well-written and clearly describes the proposed RACNN model and its motivations. However, some areas could benefit from more clarity:

1. The description of the data collection setup in Section III is lacking important details like area size, number of APs, WiFi protocol versions used, etc. This makes it difficult to fully understand the experimental scenario.
2. The architecture details in Section III could be explained more thoroughly, with proper mathematical notation for the different components like the attention layers, residual blocks, etc.
3. There could be more quantitative analysis beyond just the accuracy/error metrics, such as inference time, memory usage, etc. to better characterize RACNN's computational performance.
4. Figure 1 showing the training and testing sites and Figure 7 showing the CDF plot are extremely blurry, making it hard to perceive details. The authors should provide higher quality versions.
5. The description of the proposed RACNN architecture lacks clarity. The mathematical notation and flow from input to output is ambiguous, especially regarding how the "two channels" are processed. A more thorough explanation with equations and an architectural diagram is needed.

Experimental design

The experimental design is sound overall, with the authors evaluating on multiple datasets including a custom collected one. A few potential concerns:
1. No ablation study is performed to analyze the importance of different RACNN components like the attention layers, stacked autoencoder, etc. This would help justify the full architecture.
2. Only accuracy/error metrics are used for evaluation. It would be good to also consider other measures relevant for localization like precision, recall, robustness to environmental changes, etc.
3. Details are lacking on data preprocessing steps, how the train/test splits were done, any data augmentation used, etc.

Validity of the findings

The findings indicating RACNN's state-of-the-art performance seem valid based on the experimental results presented. However, there are some potential limitations:

1. Only small-area indoor scenarios are considered. The scalability of the approach to larger areas/buildings is not discussed.
2. No comparison is provided against non-learning based localization approaches like fingerprinting with probabilistic methods. This would give more context on RACNN's advances.
3. Practical deployment aspects like runtime performance, memory usage, power consumption etc. which are important for real-world IoT devices are not covered.

Additional comments

There are several significant weaknesses that need to be addressed first:

1. The mathematical formulation and flow of the model is ambiguous, especially how the "two channels" of CSI data are processed.
2. Proper equations, variable definitions, and an architectural diagram are needed for readers to fully comprehend the technical novelties claimed.
3. Critical figures like Figure 1 (sites) and Figure 7 (CDF plot) are extremely blurry, making it hard to interpret the results.
4. The authors need to provide higher quality versions of these figures.
5. No ablation study analyzing the importance of different RACNN components like attention layers, autoencoder, etc.
6. Lack of analysis on computational aspects like inference time, memory usage which are important for practical deployment.
7. Scalability to larger areas/buildings is not discussed.
8. Data collection setup lacking details like area size, number of APs, WiFi protocol used.
9. No information on data preprocessing steps, train/test splits, any augmentation used.
10. While the core idea and motivation for RACNN are interesting, and the results show promising performance gains, there are too many clarity issues and missing analyses in the current manuscript. Addressing the weaknesses listed above through revisions would significantly improve the quality and completeness of the work.

Reviewer 2 ·

Basic reporting

1- The dataset features should be described in more detail, including their specific characteristics and properties. Additionally, provide information on the total size of the dataset and the sizes of the train and test subsets. This information can be presented in a tabular format.

Experimental design

1. Insert pseudocode or a flowchart to outline the algorithm steps for better clarity.
2. Measure and report the time spent during the experimental results to provide insights into the computational efficiency of the proposed approach.
3. Include a Limitation section and a Discussion section to address any limitations and provide a deeper analysis of the findings.

Validity of the findings

1. Ensure that all relevant metrics are calculated and reported in the experimental results to provide a comprehensive evaluation of the proposed model's performance.
2. Provide a table that includes the parameters used for the analysis, allowing readers to understand the configuration of the model.

Additional comments

1. Discuss the cost associated with deploying the deep learning models, including the necessary hardware and software requirements.
2. Provide a detailed description of the architecture of the proposed model, outlining its components and how they contribute to the overall framework.
3. Address the accuracy/improvement percentages in both the abstract and conclusion sections, emphasizing the significance of these results.
4. Conduct a thorough proofread to eliminate any grammatical mistakes or typographical errors.
5. Enhance the clarity of the figures by improving their resolution.
6. Include a section on future work in the conclusion section if there are any potential avenues for further research.
7. Review and reference highly related research papers to improve the Related Work and Introduction sections, specifically considering the papers mentioned:
a) Privacy issues of public Wi-Fi networks
b) User awareness of privacy, reporting system, and cyberbullying on Facebook
c) Optimizing classification of diseases through language model analysis of symptoms
d) Arabic Toxic Tweet Classification: Leveraging the AraBERT Model
e) A high-quality feature selection method based on frequent and correlated items for text classification
f) A new feature selection method based on frequent and associated itemsets for text classification
g) Detecting cyberbullying using deep learning techniques: a pre-trained glove and focal loss technique
h) Developing an efficient method for automatic threshold detection based on a hybrid feature selection approach

---

## Round 0.2 · Minor Revisions

The reviewers are more towards accepting the paper with some minor suggestions. Please revise the paper and explain in a separate document what changes were carried out in response to the review. I will review the papers and changes and make a final decision.

·

Basic reporting

The manuscript is well-written overall but could benefit from clearer details on the experimental setup to help readers understand the context better. The authors have provided sufficient references, but some newer works on neural network applications in Wi-Fi-based localization could strengthen the background. Including recent research in hybrid models or alternative neural network architectures would be a useful addition.

The article follows the professional structure expected by the journal. However, some figures, particularly Figure 7 (CDF plot), are blurry and need higher resolution for better readability. The raw data is shared appropriately. The results are relevant to the hypotheses, and the article is self-contained with all necessary explanations.

Experimental design

The paper presents original research within the journal’s scope, focusing on Wi-Fi-based localization using residual attention convolutional neural networks (RACNN). The research question is well-defined and meaningful, addressing the challenge of improving localization accuracy in complex indoor environments. The authors make a clear case for how their research fills an identified gap in the field.

The investigation is carried out to a high technical standard. However, more details about the dataset collection process would help clarify how the experiments were conducted. For instance, elaborating on environmental variations such as the number of obstacles or dynamic conditions during data collection would be helpful. The methods are mostly clear, but more detailed information on the RACNN model’s architecture, including hyperparameters, would improve replicability. It would also be beneficial to include an ablation study to show how each component of the RACNN (such as attention layers and residual blocks) contributes to performance.

Validity of the findings

The findings are novel and add value to the field, particularly in demonstrating how attention mechanisms can improve localization accuracy. However, the scalability of the model to larger, more complex environments is not explored in depth and could be addressed further.

The data provided is robust, and the experiments appear statistically sound. It’s good to see precision, recall, and F1 scores included alongside accuracy. The conclusions are well-stated, linked to the original research question, and supported by the results. The authors convincingly show that their RACNN model outperforms traditional methods like K-Nearest Neighbors.

Additional comments

The paper briefly mentions computational costs like memory usage and inference time, but expanding this discussion would provide readers with a clearer understanding of the model’s feasibility in real-world applications. It would also be beneficial to discuss the limitations of the model, especially in dynamic or larger indoor environments. Providing future directions for research, such as scaling the model or optimizing it for edge devices, would enhance the overall discussion.

Reviewer 2 ·

Basic reporting

Accept.

Experimental design

-

Validity of the findings

-

Additional comments

-

---

## Round 0.3 · accepted · Accept

After reviewing the changes you made, I feel comfortable accepting your changes and moving forward. Please ensure the paper may have to be updated as per the journal's editorial staff direction. Thanks for your interest in the journal.